# Relationships between Physical and Mental Health in Adolescents from Low-Income, Rural Communities: Univariate and Multivariate Analyses

**DOI:** 10.3390/ijerph18041372

**Published:** 2021-02-03

**Authors:** Robyn Feiss, Melissa M. Pangelinan

**Affiliations:** School of Kinesiology, Auburn University, Auburn, AL 36849, USA; mgp0020@auburn.edu

**Keywords:** anxiety, body image, depressive symptoms, obesity, physical activity, self-esteem

## Abstract

Environment (i.e., rural vs. urban) and socioeconomic status (SES) are moderating factors of physical (i.e., obesity and/or physical activity) and internalizing mental health (i.e., stress, anxiety, and depressive symptoms) in adolescents. Relationships between physical and mental health have been shown in adolescents; however, research has not addressed these relationships in those from low-income, rural backgrounds. Thus, the present study characterized physical and mental health in rural, low-SES adolescents and investigated relationships between physical and mental health in this population. Data were collected from 253 10th and 11th-grade students from Title I schools in rural Alabama. Self-report measures of mental health, self-esteem, body image, and physical activity were obtained, in addition to functional fitness and physical health assessments completed at each school. Relationships between mental and physical health were assessed using Pearson correlations and multivariate data-driven cluster analysis. Positive correlations were observed between body composition and mental health symptoms, while negative correlations were observed between body image and mental health and body composition. However, sex differences were present in these relationships. The multivariate cluster analysis identified groups of individuals based on profiles of mental and physical health. This individual-level analysis identified students with greater mental and/or physical health burdens (*n* = 53 and *n* = 40) who may benefit from targeted interventions. Overall, these results provide evidence of elevated mental and physical health burdens among rural, low-income adolescents. Moreover, targeted programs are needed to provide education about the relationship between physical and mental health to reduce health burdens in both domains in this population.

## 1. Introduction

Adolescence represents a critical period during which problems in physical health (e.g., lower physical activity levels, increased obesity) and internalizing mental health (e.g., increased symptoms of depression, anxiety, stress) emerge and may lead to long-term health burdens [1]. National surveys suggest that only 26.1% of high school students report meeting physical activity guidelines (i.e., 60 min of physical activity per day) with more males (35.3%) reporting meeting guidelines than females (17.5%) [2]. However, studies employing quantitative measurement of physical activity via accelerometers report that far fewer adolescents actually meet these guidelines (i.e., 7.5% of 12- to 15-year-olds and 5.1% of 16- to 19-year-olds) [3,4]. Furthermore, the current estimates of adolescent obesity (having a BMI in the 95th percentile) range from 14.8% [2] to 20.6% [5]. These percentages have increased steadily over the past three decades [6] justifying the critical need for physical health interventions.

Current internalizing mental health statistics for adolescents are also alarming. An estimated 31.9% of adolescents ages 13–18 years have been or are currently diagnosed with an anxiety disorder [7]. Moreover, 13.3% of adolescents ages 12–17 years have experienced a major depressive episode in the last year [8], with 31.5% reporting feeling sad or hopeless almost every day for 2 or more weeks in a row [2]. Both clinical prevalence of anxiety and depression as well as reported depressive symptoms are higher in females than males [2,7,8]. Moreover, the estimated prevalence of these disorders has increased incidence by as much as 37% from 2005 to 2014 in this population [9]. Although studies have not estimated the prevalence of subclinical levels of stress or anxiety, it is likely that subclinical prevalence mirrors that of clinical diagnoses. Together, these statistics suggest that interventions during adolescence are necessary to mitigate risks for health problems across both physical and mental health domains.

Indeed, physical and internalizing mental health are inter-related across development, particularly during adolescence [10,11,12]. One conceptual model proposes that multiple neurobiological, psychosocial, and behavioral mechanisms may underlie the relationship between physical and mental health [11]. For example, changes in grey matter volume and activation, physical self-perceptions, and coping and self-regulation skills may all contribute to internalizing disorders (e.g., anxiety and depression). Moreover, multiple large cross-sectional studies of European adolescents have observed relationships between physical and internalizing mental health; lower physical activity levels were associated with greater levels of depression and anxiety [13,14], as well as worse overall well-being [14]. These findings have also been demonstrated in a longitudinal cohort study of Norwegian adolescents [15]; lower physical activity levels were associated with greater levels of depression and anxiety in male adolescents. Furthermore, physical activity programs, including participation in team sports [16] and positive youth development sport programs [17] have been associated with increased self-esteem and body size satisfaction among adolescent populations in the U.S. Although these studies provide support for the inter-relation between physical and internalizing mental health outcomes in adolescents, few studies have examined adolescents at greatest risk for physical and mental health burdens (i.e., those with limited access to medical care, and those from lower socioeconomic status (SES)).

Indeed, studies have consistently reported that environment (rural vs. urban), SES, and sex are moderating factors of physical health outcomes in adolescents. Specifically, females [2], adolescents from rural areas [18], and those from families with lower SES [19] report lower physical activity levels and are more likely to be overweight or obese, compared with males, adolescents from urban areas, or those from families with higher SES, respectively.

Evidence regarding the impact of SES on mental health status has been mixed, possibly due to the fact that it may interact with factors, such as race, to create individualized experiences and perspectives. Indeed, intersectionality theory proposes that demographic factors, including sex, race/ethnicity, SES, and culture interact to form individualized experiences and perspectives which contribute to differences in mental health attitudes, access, and service use [20,21,22]. One literature review suggested that low SES was associated with greater self-reported internalizing mental health problems among adolescents [23], however, data from a national survey found no relationship between SES and internalizing diagnoses [7]. These divergent findings may be related to the way in which mental health problems were operationalized (i.e., self-report vs. clinical diagnoses). Additionally, studies investigating the impact of environment on mental health, suggest that there are greater barriers for receiving mental health services in rural environments, for example, cultural perception, ability to travel to/from services, and quality of care [24,25]. Collectively, along with intersectionality theory, these studies suggest that adolescents from rural areas and low SES backgrounds are at the greatest risk for cumulative physical and mental health burden and that these problems may underlie the health disparities reported in adults including metabolic disease, cardiovascular disease, and mental health disorders. Moreover, despite the well-established relationship between physical and internalizing mental health outcomes in other populations of adolescents, no study has examined this relationship in rural, low-income communities.

### Current Study

The overarching purpose of this study was to quantify physical and internalizing mental health status among students attending rural Title I schools (i.e., schools that receive supplemental federal funding due to high concentrations of low-income students). We hypothesized that this population would exhibit a high prevalence of physical and internalizing mental health problems (i.e., obesity, anxiety, and depressive symptoms) compared to national standards. We also hypothesized that better physical health outcomes (i.e., lower body fat percentage and BMI) and greater physical activity would be related to fewer internalizing mental health symptoms. We hypothesized that this cumulative relationship may be moderated by sex, such that females would report lower physical activity and more mental health symptoms than males resulting in sex differences in the relationship between physical and mental health. Lastly, using a data-driven analytic approach, we aimed to identify clusters of students based on profiles of physical and mental health. For this exploratory analysis, four clusters of adolescents were hypothesized: those with good mental and physical health, those with good mental but poor physical health, those with poor mental but good physical health, and those with poor mental and physical health. These clusters could be used to identify students that may benefit from targeted interventions to mitigate elevated physical and/or mental health burdens.

## 2. Materials and Methods

### 2.1. Participants

Tenth and eleventh grade students were recruited from four Title I high schools in rural Alabama. Title I schools receive supplemental federal funding due to high concentration of students (i.e., at least 40% of students) from low-income families that receive free or reduced meals. Both counties in which the high schools were located are considered rural, with an Index of Relative Rurality (IRR) of 0.5 (Chambers (IRR = 0.5); Tallapoosa (IRR = 0.5)) in 2010 [26] Both counties are primarily White, with Tallapoosa county (70.0% White, 27.4% Black) having a higher percentage of White residents than Chambers county (57.7% White, 39.6% Black) [27].

### 2.2. Measures

#### 2.2.1. Parent Demographic Survey

The parent demographic survey consisted of four questions to determine the ethnicity/race of the parent, their household income, their level of education, and their relationship to the child.

#### 2.2.2. Student Demographic Survey

The student demographic survey was a 3-item questionnaire used to determine the participant’s age, sex, ethnicity/race.

#### 2.2.3. PROMIS Pediatric Item Bank v1.0-Physical Activity-Short Form 8a

This questionnaire is an 8-item measure of the frequency of moderate-to-vigorous physical activity (MVPA) per week. The conceptual framework and item development were a part of the National Institute of Health’s Patient Reported Outcome Measurement Information System (PROMIS) [28,29,30,31] and the measure is appropriate for use in the general population [29,30]. This physical activity measure was found to be highly reliable among this study sample (α = 0.94). Scores from each question were summed to form a composite score ranging from 8 (did not exercise at all) to 40 (participated in MVPA 6 or 7 days per week).

#### 2.2.4. PROMIS Pediatric Item Bank v2.0-Anxiety-Short Form 8a

This questionnaire is an 8-item measure of anxiety symptoms (e.g., focusing on worrying, nervousness, and fear) and is recommended for use in the general population [32]. This anxiety measure was found to be highly reliable among this study sample (α = 0.91). Scores from each question were summed to calculate a composite score ranging from 8 to 40, with higher scores representing higher levels of anxiety symptoms.

#### 2.2.5. PROMIS Pediatric Item Bank v2.0-Depressive Symptoms-Short Form 8a

This questionnaire is an 8-item measure of depressive symptoms, focusing on common depressive symptoms among children and adolescents [33] and is appropriate for use in the general population [32]. This depressive symptoms measure was found to be highly reliable among this study sample (α = 0.93). Scores from each question were summed to calculate a composite score ranging from 8 to 40, with higher scores representing higher levels of depressive symptoms.

#### 2.2.6. PROMIS Pediatric Item Bank v1.0-Psychological Stress Experiences-Short Form 8a

This questionnaire is an 8-item measure of psychological symptoms commonly associated with stress and is recommended for use in the general population [34,35]. This psychological stress measure was found to be highly reliable among this study sample (α = 0.91). Scores from each question were summed to calculate a composite score ranging from 8 to 40, with higher scores representing higher levels of psychological stress symptoms.

#### 2.2.7. Body Shape Satisfaction Scale (BSS)

A modified version of the Body Shape Satisfaction Scale [36] was used to assess body satisfaction/dissatisfaction. This 13-item questionnaire has been used with both male and female adolescents [37]. This body satisfaction measure was found to be highly reliable among this study sample (α = 0.92). Scores from each item were summed to calculate a composite score ranging from 13 to 91, with higher scores representing greater body satisfaction.

#### 2.2.8. Rosenberg Self-Esteem Scale (RSES)

The Rosenberg Self-Esteem Scale short form [38] was used to assess global self-esteem. This 6-item questionnaire has been used with adolescents with varying demographic backgrounds [39]. This self-esteem measure was found to be reliable among this study sample (α = 0.83). Scores from each item were combined to calculate a composite score ranging from 6 to 24, with higher scores representing greater self-esteem.

#### 2.2.9. Functional Fitness and Health Testing

FITNESSGRAM^®^ physical fitness testing consisted of anthropometrics (height, weight, body mass index (BMI)), body composition (% fat mass, % lean mass), resting heart rate, blood pressure, muscular strength and endurance (push-ups, curl-ups), and aerobic capacity (Progressive Aerobic Cardiovascular Endurance Run (PACER)). Body composition was measured using a TANITA total body composition analyzer (SC-331S Total Body Composition Analyzer, TANITA) and blood pressure and heart rate were measured via an Omron^®^ automatic blood pressure monitor (5 Series Upper Arm Blood Pressure Monitor BP742N, Omron Healthcare).

### 2.3. Procedure

The study was conducted in accordance with the Declaration of Helsinki, and the protocol was approved by the Institutional Review Board at Auburn University (Protocol #: 18-109 MR 1803). Data collection took place each of the Title I high schools (≈3 h per school). Information packets with details about the study were distributed to students 1–2 weeks prior to data collection. Out of 575 students, 253 students participated in the study (44%). Prior to data collection, parents completed the informed consent to allow their child to participate and the parent demographic survey. Students also completed the assent form prior to their participation in the study. Students completed the student demographic survey and the PROMIS Pediatric Physical Activity, PROMIS Pediatric Anxiety, and PROMIS Pediatric Depression questionnaires prior to the in-school data collection. Students completed the PROMIS Pediatric Psychological Stress questionnaire, the BSS, and the RSES during the in-school program. Groups of 4–8 students of the same sex rotated through a circuit of the tasks for the functional fitness and physical health assessment. Data were collected by trained research assistants during a single visit to each school (≈3 h total).

### 2.4. Statistical Analyses

Rates of anxiety, depression, and stress were computed based on the PROMIS questionnaires. Overweight and obesity rates were calculated using the measured BMI according to Center for Disease Control (CDC) standards (overweight is 85–95th percentile and obese is ≥95th percentile). Body fat percentage was categorized into very lean, healthy fitness zone (HFZ), needs improvement (NI), and needs improvement-health risk (NI-HR) according to FITNESSGRAM^®^ standards. Physical fitness scores (push-ups, curl-ups, PACER) were categorized into HFZ, NI, and NI-HR (when applicable) according to FITNESSGRAM^®^ standards. Self-report physical activity, anxiety and depression scores were standardized and categorized into normal, mild, moderate, and severe according to PROMIS norms [40,41]. Psychological stress scores were standardized and categorized into low, average, high or very high according to PROMIS norms [40,41].

MATLAB version R2018a (MathWorks Inc., Natick, MA, USA) was used to conduct independent samples t-tests to examine sex differences in physical activity and prevalence internalizing mental health symptoms. Additionally, we conducted Pearson correlations, separated by sex, to examine bivariate correlations amongst the physical (body fat percentage, BMI, physical activity) and mental health (stress, depression, anxiety, body image, self-esteem) dependent measures as well as possible sex differences in these relationships. The three physical health measures were selected as they provide an overall representation of physical health. The level of significance was set to *p* < 0.05.

In addition to these standard analyses, a multivariate data-driven cluster analysis (using the clustergram function in MATLAB version R2018a (MathWorks Inc., Natick, MA, USA)) was used to identify groups of students based on 13 variables representing internalizing mental health (stress, depression, anxiety), physical health characteristics (physical activity level, body fat percentage, BMI, systolic blood pressure, diastolic blood pressure, resting heart rate, number of sports), and physical fitness measures (push-ups, curl-ups, PACER). The cluster analysis normalized each variable and computed the Euclidian distance between individuals based on the normalized score for all 13 variables. Individuals and clusters were linked based on the shortest average distance between individuals. The cluster analysis also groups variables based on the similarity between the profile of data across individuals. Unlike k-means clustering, the number of clusters is not predetermined in hierarchical divisive cluster analysis. Therefore, the number of clusters is based on the distance function, linkage function, and the structure of the dataset. However, variance in cluster output due to the selection of the distance function and linkage functions is minimized when data are normalized.

### 2.5. Missing Data

As participants completed the consent forms and some of the questionnaires prior to the in-school data collection, including the demographic survey, PROMIS Pediatric Physical Activity, PROMIS Pediatric Anxiety, and PROMIS Pediatric Depression questionnaires, data are missing at random for the following reasons: the student completed and returned the questionnaire packet prior in-school data collection, but was absent for in-school data collection; the student was present for the in-school data collection, but did not return or complete the questionnaire packet; the student was present for the in-school data collection, but chose not to complete a test or questionnaire; PROMIS Psychological Stress questionnaire was not distributed to a student due to miscommunication. Lastly, as the BSS and RSES were added to the protocol after some schools had already completed data collection, only a subset of students completed these questionnaires.

## 3. Results

A total of 253 adolescents, ages 15- to 18-years participated in the study. Participant demographics are presented in Table 1.

### 3.1. Mental and Physical Health Characteristics

Less than half the participants were in the normal range for anxiety (43.5%), depressive (43.5%), and stress (43.9%) symptom levels. Specifically, 36.4% reported moderate or severe anxiety symptoms, 35.9% reported moderate or severe depressive symptoms, and 28.0% reported high or very high psychological stress symptoms (see Table 2). Females exhibited significantly lower physical activity (*t*(228) = −4.35, *p* < 0.001) and greater anxiety (*t*(232) = 5.90, *p* < 0.001), depressive (*t*(233) = 5.35, *p* < 0.001), and stress symptoms (*t*(180) = 5.96, *p* < 0.001) compared to males.

Based on BMI measurements, 39.1% of participants were either overweight (15.8%) or obese (23.3%). Almost half of the participants were categorized as having healthy body composition (47.8% for body fat percentage and 47.4% for BMI) according to FITNESSGRAM^®^ standards. About half (49.4%) met FITNESSGRAM^®^ standards for push-ups and curl ups (49.8%). Participants also performed poorly on the PACER test, with only 20 participants (7.9%) meeting criteria for healthy standards, while the majority (63.2%) fell into the NI-HR category (see Table 3). Although many students did not perform well on the physical fitness tests, 64.8% of students fell into the normal range of physical activity based on the PROMIS questionnaire. The distributions with category cut-points for the PROMIS physical activity and mental health T-scores are depicted in Figure 1.

### 3.2. Relationships between Mental and Physical Health

The bivariate correlations revealed significant positive correlations between body fat percentage and anxiety (*r_m_* = 0.22, *p_m_* = 0.04; *r_f_* = 0.19, *p_f_* = 0.04) among males and females and higher stress among males (*r* = 0.23, *p* = 0.02), higher body fat percentage was associated with more anxiety symptoms for males and females and higher stress for males. Additionally, body fat percentage was negatively correlated with body satisfaction for females (*r* = −0.45, *p* = 0.02), but not males (*r* = −0.26, *p* = 0.09), higher body fat percentage was associated with lower body satisfaction among females, but not males. BMI was positively correlated with anxiety (*r_m_* = 0.26, *p_m_* = 0.02; *r_f_* = 0.21, *p_f_* = 0.02) and depression (*r_m_* = 0.22, *p_m_* < 0.05; *r_f_* = 0.18, *p_f_* = 0.04) among males and females and stress among males (*r* = −0.26, *p* = 0.03), higher BMI was associated with more anxiety and depressive symptoms among males and females and higher stress among males. BMI was also negatively correlated with body satisfaction (*r_m_* = −0.31, *p_m_* = 0.02; *r_f_* = −0.45, *p_f_* = 0.02) among males and females and self-esteem among females (*r* = −0.32, *p* = 0.02), higher BMI was associated with lower body satisfaction among males and females and lower self-esteem among females. Body satisfaction was positively correlated with self-esteem (*r_m_* = 0.36, *p_m_* = 0.02; *r_f_* = 0.49, *p_f_* = 0.02;) and negatively correlated with depression (*r_m_* = −0.31, *p_m_* = 0.04; *r_f_* = −0.40, *p _f_* = 0.04) among males and females and anxiety among females (*r* = −0.44, *p* = 0.02), higher body satisfaction was associated with greater self-esteem and less depressive symptoms among males and females and less anxiety symptoms among females. Self-esteem was also negatively correlated with anxiety (*r_m_* = −0.44, *p_m_* < 0.01; *r_f_* = −0.44, *p_f_* < 0.01), depression (*r_m_* = −0.49, *p_m_* < 0.001; *r_f_* = −0.71, *p_f_* < 0.001) and psychological stress (*r_m_* = −0.39, *p_m_* < 0.01; *r_f_* = −0.64, *p_f_* < 0.01), higher self-esteem was associated with less internalizing mental health symptoms for both males and females. Physical activity was positively correlated with anxiety (*r* = 0.18, *p* < 0.05) and depression (*r* = 0.24, *p* < 0.01) among females, but not males, indicating greater physical activity was associated with increased anxiety and depressive symptoms among females. Interestingly, neither body fat percentage (*r_m_* = −0.03, *p_m_* = 0.78; *r_f_* = −0.14, *p_f_* = 0.13) nor BMI (*r_m_* = −0.10, *p_m_* = 0.35; *r_f_* = −0.12, *p_f_* = 0.19) were significantly correlated with self-report physical activity levels among males or females. A correlation matrix containing relationships between all the examined variables for male participants is depicted in Table 4 and females are in Table 5.

### 3.3. Data Driven Classification of Variable Clusters

Figure 2 depicts the cluster results. The first-level separation of variables in the cluster analyses (top of Figure 2) split the physical fitness (i.e., curl-up, push-up, PACER) and physical activity measures (i.e., number of sports, self-report physical activity) from the blood pressure, resting heart rate, body composition (i.e., percent body fat, BMI), and internalizing mental health measures (i.e., anxiety, depression, stress). This first level clustering suggests that body composition and resting heart rate are more closely related to mental health than physical fitness, supporting the use of body composition measures to explore relationships between physical and mental health.

### 3.4. Data Driven Classification of Participant Clusters 

The optimal number of clusters was examined using the NbClust package in R/R-Studio (version 1.2.1335). This package evaluates 30 cluster indicies and based on majority rule, the optimal number of clusters is identified. Four was identified as the optimal number of clusters. The cluster analysis revealed four distinct clusters of participants as well as three separate branches each representing 1–5 individuals that were distinct from the other clusters (Figure 2 left side). The first branch separated one female whose data profile was distinct from other in that she exhibited very high body fat percentage, BMI, anxiety, depression, and high curl-up score (top). The second branch separated two individuals, one male and one female, with very high blood pressure and slightly above average body composition (bottom). The third branch separated one male with very high body fat percentage, BMI, systolic blood pressure, and curl-up scores.

With respect to the larger clusters of individuals (i.e., those with more than five participants), Cluster 1 (blue) included 31 males and 2 females. The profile for the participants in this cluster included those with a high number of sports, healthy body composition, above average physical fitness scores, and mostly normal or mild levels of anxiety, depression, and stress. Specifically, only 7 out of 33 exhibited moderate mental health symptoms. Cluster 2 (purple) included 106 participants (49 males, 57 females) who had healthy body composition and typically met standards for the push-ups (78/106 (73.6%) HFZ) and curl-ups (71/106 (67.0%) HFZ), but not the PACER test (3/106 (2.8%) HFZ). These participants also reported generally normal or mild levels of anxiety, depressive, or stress symptoms with 34 out of 106 participants exhibiting moderate or severe symptoms). Overall, the majority of the individuals in Clusters 1 and 2 exhibited few physical and mental health problems.

Clusters 3 (*n* = 53) and 4 (*n* = 40) identified participants experiencing greater physical and/or mental health problems and included a large number of female students. Cluster 3 (red) included 11 males and 42 females with mixed body composition (21 out of 53 with normal BMI) and performance on the physical fitness tasks (31 out of 53 in the healthy fitness zone). Additionally, all but one individual reported moderate or severe anxiety, depressive, and/or stress symptoms. Cluster 4 (green) included 40 participants (8 males, 32 females) with unhealthy body composition and poor physical fitness performance (needs improvement or needs improvement health risk). Compared to Clusters 1 and 2, anxiety, depression and stress levels were elevated, with 27 of the 40 participants reporting moderate or severe anxiety, depressive, and/or stress symptoms.

## 4. Discussion

### 4.1. Prevalence of Physical and Mental Health Issues

The combination of standard statistical analyses and a data-driven multivariate analysis allowed for unique insights at the group and individual levels, respectively. At the group level, these data indicate that rural, low-income adolescents exhibited greater obesity and internalizing mental health issues than previous state [42] and national-level data noted [7,42]. Specifically, the obesity rate of 26.5% in this sample was 8.3% greater than the most recently reported obesity rate of adolescents in the state of Alabama (18.2%) [42] and 11.7% higher than the national average (14.8%) [2]. These data are alarming in that Alabama was ranked 42 out of 51 (including all 50 states and the District of Columbia) for adolescent obesity [42] and yet, a greater percentage of participants in this study were considered overweight or obese compared to the most recent report. The mental health statistics were also higher than previous national estimates; 36.4% reported moderate or severe anxiety symptoms, 36.0% reported moderate or severe depressive symptoms, and 28.1% reported high or very high psychological stress symptoms. A previous national estimate for adolescent anxiety was 4.5% lower (31.9%) [7] and that for depression was 6% lower (30%) [43] than the current sample. Alarmingly, less than half the participants were in the normal range for anxiety (43.5%), depressive (43.5%), and stress (43.9%) symptom levels. With respect to the mental health prevalence, the present results replicate and extend previous research suggesting that low-income, rural populations may be at elevated risk for mental health issues [23,24,25], with a large percentage of participants exhibiting moderate or severe symptoms of depression, anxiety, or stress. Moreover, consistent with national-level data, females in this population exhibited a significantly greater risk for both mental [2,7,43] and physical [2] health issues than males.

### 4.2. Relationships between Mental and Physical Health

The bivariate correlations revealed that higher body composition was associated with greater anxiety and depressive symptoms among males and females and greater stress among males. Relationships between body composition, body satisfaction, and self-esteem differed by sex, where body composition was more closely related to body satisfaction and self-esteem in females than males. Additionally, higher self-esteem was associated with less internalizing mental health symptoms among both males and females. Unexpectedly, there was a positive relationship between physical activity and anxiety depressive symptoms among females. The cause of this relationship is unknown and should be explored in future research. The present analysis demonstrated that relationships between body composition, body satisfaction, and mental health outcomes are different for males and females. This is in line with previous research which found that the relationships between body image and mental health outcomes were sex dependent; males that rated themselves as thin/very thin and females that rated themselves as very fat/chubby or thin/very thin had a higher risk of symptoms of anxiety or depression [15]. However, that study did not find a relationship between BMI and mental health outcomes [15]. Future studies are needed to determine if the relationship between mental and physical health is mediated by body composition or body satisfaction. To our knowledge, this was the first study to address relationships between physical health and mental health among low-income, rural adolescents. Therefore, additional studies are needed to replicate and extend the present study in similar populations.

At the variable level, the cluster analysis found that the body composition measures (body fat percentage, BMI) and resting heart rate were more closely related to mental health compared to the physical activity or physical fitness measures. This is somewhat in contrast with findings from Fløtnes et al. (2011) [15] that greater physical activity was related to lower anxiety/depressive symptoms in adolescent males, but that there was no relationship between BMI and anxiety/depressive symptoms for males or females. These discrepancies may be due to the population studied (Norwegian adolescents vs. rural, low-income, American adolescents), particularly given the high prevalence of obesity among the present sample.

At the individual level, the cluster analysis identified groups of participants with similar physical and mental health profiles. This analysis revealed four main clusters. The first two clusters identified students with average to good physical and mental health. A third cluster identified individuals with poor mental health but average to good physical health, while the fourth cluster identified individuals with poor physical health and poor to average mental health. While these clusters were not entirely in line with the hypothesized clusters (i.e., good physical and mental health, poor physical and mental health, good physical and poor mental health, poor physical and good mental health), the cluster analysis did reveal groups of individuals with greater mental and physical health burdens. These individuals may benefit from targeted interventions to reduce physical and/or mental health burdens. It is important to note that the cluster analysis normalizes each individual’s data, such that those who exhibit an unhealthy profile are unhealthy compared to the rest of the sample. Given that this population is already at high-risk, those identified by the cluster analysis can be considered in greatest need for intervention. Data acquired from similar populations may yield similar clusters and identify those at highest risk.

### 4.3. Limitations and Future Research

This study specifically targeted high school students from rural, Title I schools as an important and under-represented group in the research literature. As such, this sample lacked diversity with respect to race and socioeconomic status; nearly all students were African American/Black (45.1%) or Caucasian/White (41.5%), and nearly half (43.1%) reported household incomes less than USD 30,000. The present categories for income were based on those used a previous study [44] and the median income for the state of Alabama. As such, the very lowest income category did not sufficiently differentiate those in the very low-income category (e.g., USD < 30,000). Future studies in this population may need additional income stratification for those with low income (e.g., USD < 15,000, USD 15,000–29,999, USD 30,000–49,999). Additional future research addressing physical and mental health burdens should focus on comparing urban and rural adolescents with similar SES and racial backgrounds. This comparison would help to determine which of the three demographic factors (i.e., environment, race, or SES) is the most influential. For example, if minimal differences are found between urban and rural adolescents, this may suggest that race and/or SES are the more influential factors. Additionally, it was not possible to conduct multilevel modeling to assess school-level differences as only four schools of varying size were included in the present study. Future studies are necessary to replicate and extend these findings in a larger population of adolescents from rural, Title I schools to enable multilevel analyses. Further, a subjective, self-report measure of physical activity was used for the study, which may result in overestimation of physical activity. Future studies should strive to use an objective measure of physical activity such as accelerometry. Lastly, this was a cross-sectional study whose aim was to determine health disparities in rural, low-income schools and demonstrate the utility of data-driven methods to identify groups of individuals that may benefit from targeted health interventions. Therefore, this study was not able to provide insights regarding changes in physical and mental health outcomes in this population or the efficacy of targeted interventions with this population. With that said, targeted interventions such as school-based physical activity and/or mental health programs and longitudinal tracking are essential to reduce mental health and physical health burdens, particularly in female adolescents in this population.

### 4.4. Implications for Policy, Theory, and Practice

As the prevalence of both physical and internalizing mental health problems continues to increase among adolescents from this population [2,18,19,43], local education and health policies are needed to detect, treat, and prevent physical and mental health problems. The efficacy of school-based mental and physical health interventions has been demonstrated previously [45,46,47] and may serve as a convenient method of targeting this unique population, especially if there are barriers to access to specialized health programs (e.g., transportation, limited facilities, cost, etc.). Moreover, programs delivered at the school may also help reduce barriers to physical activity participation as the programs could enable time during the school day to be physically active. Culturally sensitive practices and incorporating group sessions of same-race participants may be particularly important for minority, low-income, and/or rural populations [48,49].

State-level policies requiring schools to provide mental health education, such as those implemented by New York, Virginia, and Florida, may improve mental health outcomes, including reducing mental health stigma known to limit help-seeking behaviors [50,51]. Broad dissemination of similar programs, particularly in the rural South, may reduce mental health burdens that continue into adulthood. Programs that target mental and physical health burdens by including both physical activity and psychotherapy components have been shown to increase physical activity and reduce mental health symptoms [52]. Education and health policies should continue to integrate physical activity and mental health education in school curricula.

While the present findings do not speak to proposed mechanisms underlying the relationship between physical and mental health (i.e., neurobiological, psychosocial, or behavioral), sex was identified as one individual factor that affects physical activity and serves as a moderator of the relationship between physical and mental health. These findings are consistent with the intersectionality paradigm, which suggests that sex, race, SES, and environmental factors interact to create individual experiences, perspectives, and behaviors [20,21]. For example, although two adolescents may both live in a rural area or have a similar household income, their experiences, perspectives, and behaviors may differ based on their race. Indeed, intersectionality has been used as a framework for examining interactions amongst factors associated with health disparities in minority populations [53,54,55]. However, this framework has not been used in previous studies examining relationships between mental and physical health among adolescents. The present findings also have practical implications for clinicians and researchers. First, individuals in rural, low SES populations are at an increased risk for both physical and mental health problems, therefore these populations should be targeted for future interventions. Additionally, these types of individual level analyses, may help identify individuals in a sample who have the greatest need for intervention.

## 5. Conclusions

The current study addressed several knowledge gaps. First, elevated physical and internalizing mental health burdens were observed in a group of adolescents from rural, low-income backgrounds. Second, relationships between mental and physical health were identified at the group and individual levels in this population. Overall, an elevated physical and mental health burden was indeed evident, particularly amongst female adolescents. Moreover, the data-driven cluster analysis enabled the identification of individuals at greatest risk based on profiles of physical and mental health, which may be useful for future interventions. Indeed, targeted programs are needed to provide education about the relationship between physical and mental health to reduce health burdens in both domains in this at-risk population.

## Figures and Tables

**Figure 1 ijerph-18-01372-f001:**
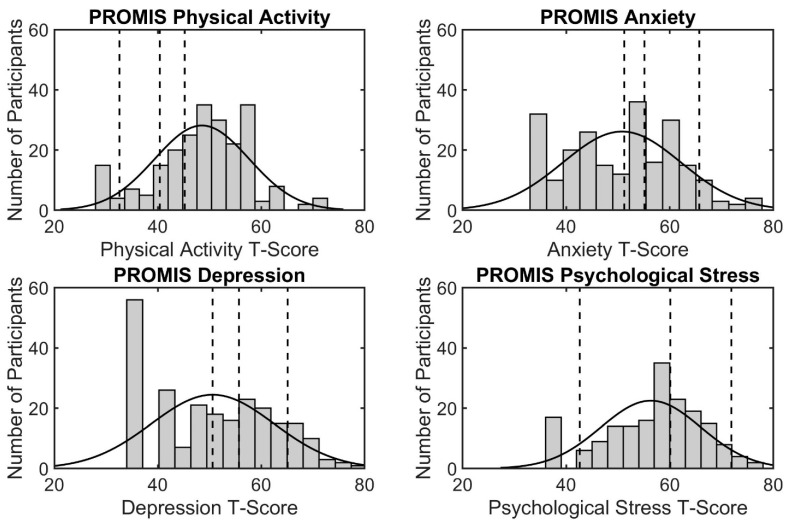
Distribution of T-scores for each of the four PROMIS measures. For each figure, the X axis represents PROMIS standardized T-score and the X-axis represents the number of participants. The vertical dashed lines depict the cut-points for each category (normal, mild, moderate, severe (physical activity, anxiety, depression) or low, average, high, very high (psychological stress)).

**Figure 2 ijerph-18-01372-f002:**
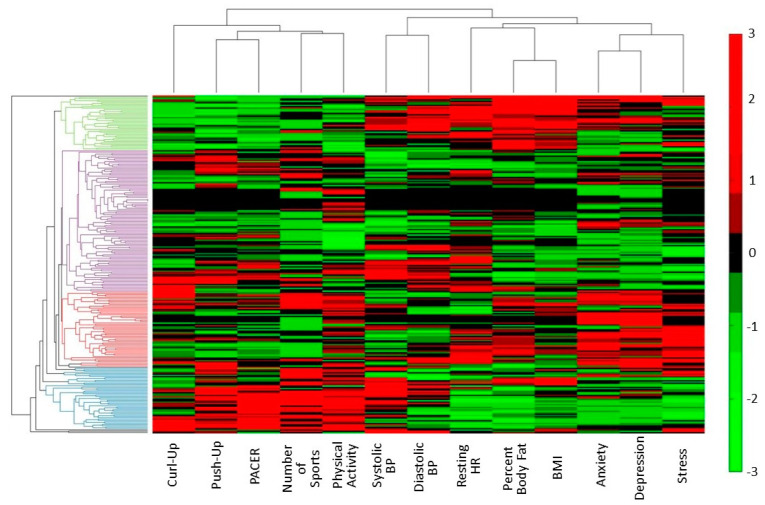
Clustergram of the population across the 13 variables of interest. The X-axis represents each of the 13 variables of interest and their clustering based on similar response profiles across participants. The Y-axis represents each participant and their clustering based on similar profiles across all 13 variables (i.e., shortest Euclidian distance across all variables). The red-green color coding represents the normalized scores for each variable, where red represents high scores for a particular variable, black represents average scores, and green represents low scores.

**Table 1 ijerph-18-01372-t001:** Participant demographics.

Demographic Variables	Number of Participants (%), N = 253
Sex	
Male	114 (45.1%)
Female	139 (54.9%)
Age	
15	151 (59.7%)
16	80 (31.6%)
17	15 (5.9%)
186	1 (0.4%)
Not Reported	6 (2.4%)
Ethnicity	
Caucasian/White	105 (41.5%)
African American/Black	114 (45.1%)
Hispanic/Latino	4 (1.6%)
Asian/Pacific Islander	2 (0.8%)
Native American	1 (0.4%)
Mixed Race	11 (4.3%)
Not Reported	16 (6.3%)
Household Income	
USD <30,000	109 (43.1%)
USD 30,000–49,999	40 (15.8%)
USD 50,000–99,999	53 (20.9%)
USD 100,000+	20 (7.9%)
Not Reported	31 (12.3%)
Parental Education	
Did Not Finish HS	39 (15.4%)
HS Diploma/GED ^1^	74 (29.2%)
Some College	44 (17.4%)
Trade, Technical, or Vocational Training	11 (4.3%)
Associate degree	26 (10.3%)
Bachelor’s Degree	26 (10.3%)
Master’s Degree	10 (4.0%)
Professional Degree	4 (1.6%)
Doctoral Degree	0 (0.0%)
Multiple Degrees Selected	4 (1.6%)
Not Reported	15 (5.9%)

^1^ GED = General Education Development.

**Table 2 ijerph-18-01372-t002:** PROMIS ^1^ categorization.

Questionnaire	Normal	Mild	Moderate	Severe
PROMIS Physical Activity	164 (64.8%)	32 (12.6%)	19 (7.5%)	15 (5.9%)
PROMIS Anxiety	110 (43.5%)	32 (12.6%)	70 (27.7%)	22 (8.7%)
PROMIS Depression	110 (43.5%)	34 (13.4%)	58 (22.9%)	33 (13.0%)
	Low	Average	High	Very High
PROMIS Psychological Stress	17 (6.7%)	94 (37.2%)	60 (23.7%)	11 (4.3%)

^1^ PROMIS = Patient Reported Outcome Measurement Information System.

**Table 3 ijerph-18-01372-t003:** FITNESSGRAM categorization.

Test	Underweight	Normal	Overweight	Obese
Obesity (by BMI)	4 (1.6%)	120 (47.4%)	40 (15.8%)	59 (23.3%)
	Very Lean	HFZ	NI	NI–HR ^1^
FITNESSGRAM—Body Fat	6 (2.4%)	121 (47.8%)	51 (20.2%)	44 (17.4%)
		HFZ	NI	NI–HR ^1^
FITNESSGRAM—PACER ^2^		20 (7.9%)	39 (15.4%)	160 (63.2%)
		HFZ	NI	
FITNESSGRAM—Push-Up		125 (49.4%)	98 (38.7%)	
FITNESSGRAM—Curl-Up		126 (49.8%)	98 (38.7%)	

^1^ NI–HR = Needs Improvement–Health Risk; NI = Needs Improvement; HFZ = Healthy Fitness Zone; ^2^ PACER = Progressive Aerobic Cardiovascular Endurance Run.

**Table 4 ijerph-18-01372-t004:** Pearson correlations between body composition, physical activity, mental health, body satisfaction, and self-esteem for males.

	Body Fat %	BMI	Physical Activity	Anxiety	Depressive Symptoms	Psychological Stress	Body Satisfaction	Self-Esteem
Body Fat %	1							
BMI	0.899 ***	1						
Physical Activity	−0.031	0.103	1					
Anxiety	0.217 *	0.257 *	0.013	1				
Depressive Symptoms	0.149	0.217 *	0.103	0.640 ***	1			
Psychological Stress	0.232 *	0.223 *	−0.013	0.441 ***	0.555 ***	1		
Body Satisfaction	−0.258	−0.305 *	−0.100	−0.122	−0.307 *	−0.065	1	
Self-Esteem	−0.206	−0.236	0.110	−0.441 **	−0.490 **	−0.392 **	0.360 *	1

*** *p* < 0.001; ** *p* < 0.01; * *p* < 0.05.

**Table 5 ijerph-18-01372-t005:** Pearson correlations between body composition, physical activity, mental health, body satisfaction, and self-esteem for females.

	Body Fat %	BMI	Physical Activity	Anxiety	Depressive Symptoms	Psychological Stress	Body Satisfaction	Self-Esteem
Body Fat %	1							
BMI	0.928 ***	1						
Physical Activity	−0.140	−0.121	1					
Anxiety	0.188 *	0.209 *	0.178 *	1				
Depressive Symptoms	0.148	0.181 *	0.243 **	0711 ***	1			
Psychological Stress	0.163	0.154	0.178	0.526 ***	0.566 ***	1		
Body Satisfaction	−0.445 *	−0.456 *	0.166	−0.440 *	−0.407 *	0.018	1	
Self-Esteem	−0.182	−0.320*	−0.112	−0.436 **	−0.709 ***	−0.635 **	0.491 *	1

*** *p* < 0.001; ** *p* < 0.01; * *p* < 0.05.

## Data Availability

The data presented in this study are available on request from the corresponding author. The data are not publicly available due to privacy concerns.

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
