# Peer review of "Relationships between Physical and Mental Health in Adolescents from Low-Income, Rural Communities: Univariate and Multivariate Analyses"

_ijerph, 2021, doi:10.3390/ijerph18041372_

Round 1

Reviewer 1 Report

The authors examined the prevalence and correlates of physical and mental health among rural, low-income adolescents, and conducted further cluster analysis. In this population, the prevalence of physical and mental health burdens was higher than in the general population, correlates of physical and mental health were found, and clusters requiring intervention were identified. While this study provides important evidence, there are some concerns that need to be addressed.

Major Comments

  1. Results of the abstract (Line 24 on page1)

Please add the number of people in cluster 3 and cluster 4 in the cluster analysis.

  1. Results

Please add the results of the correlation between depression and physical activity to the text.

  1. Discussion (Line 336 on page12)

The positive correlation between depression and physical activity could be due to the distribution of depression being skewed to the right. Present the scatter plot and try the Spearman's rank sum test.

  1. Discussion (Line 329 and 338 on page12; Line 433 on page14)

In the discussion, the gender differences in the prevalence of mental and physical health burdens and the relationship between mental and physical health are mentioned, the data should be presented. Please present the prevalence of mental and physical health burdens by gender and the correlations by gender. It would allow for a more in-depth discussion.

  1. Discussion (Line 356 on page12)

The physical activity in this study is self-reported PA, which indicates subjective PA. Therefore, the prevalence of PA in this study may be overestimated. An objective evaluation using an accelerometer is desirable. Please add this point. 

Minor Comments

  1. Title 

The title misleads people into thinking that you have done a multi-level analysis. Please change it.

  1. Method in Abstract

Add that it was adjusted for gender in the correlation analysis.

  1. Background (Line 83 on page2)

Please add a brief explanation of the intersectionality theory.

  1. Method (Line 206 on page6)

Add that it was adjusted for gender in the correlation analysis.

  1. Results (Line 238 on page6)

The values are different from the table. Make the denominator consistent.

  1. Tables

Spell out the abbreviations in the foot note.

Table 1 HS GED

Table 3 PACER

Author Response

The authors examined the prevalence and correlates of physical and mental health among rural, low-income adolescents, and conducted further cluster analysis. In this population, the prevalence of physical and mental health burdens was higher than in the general population, correlates of physical and mental health were found, and clusters requiring intervention were identified. While this study provides important evidence, there are some concerns that need to be addressed.

Major Comments

Results of the abstract (Line 24 on page1)

  1. Please add the number of people in cluster 3 and cluster 4 in the cluster analysis.

This change has been made. A total of 53 participants were in cluster 3 and 40 in cluster 4.

Results

  1. Please add the results of the correlation between depression and physical activity to the text.

The following text has been added (pg 8, section 3.2, lines 286-288): “Physical activity was positively correlated with anxiety (r = .18, p < .05) and depression (r = .24, p < .01) among females, but not males, indicating greater physical activity was associated with increased anxiety and depressive symptoms among females.”

Discussion

  1. (Line 336 on page 12) The positive correlation between depression and physical activity could be due to the distribution of depression being skewed to the right. Present the scatter plot and try the Spearman's rank sum test.

The Spearman rank sum test results (r = .19, p = .003), were nearly identical to the original Pearson correlation analyses (r = .18, p = .006). Therefore, we kept the original analysis with Pearson correlations.

  1. (Line 329 and 338 on page12; Line 433 on page14) In the discussion, the gender differences in the prevalence of mental and physical health burdens and the relationship between mental and physical health are mentioned, the data should be presented. Please present the prevalence of mental and physical health burdens by gender and the correlations by gender. It would allow for a more in-depth discussion.

We have added independent t-tests to assess sex differences in physical activity and prevalence of internalizing mental health symptoms. Additionally, the correlation analyses have been changed to assess correlations among males and female separately to allow a more in-depth exploration of sex differences in these relationships. These changes to the statistical analyses are described on pg 5, section 2.4, lines 199-204:

“MATLAB version R2018a (MathWorks Inc., Natick, MA, USA) was used to conduct independent samples t-tests to examine sex differences in physical activity and prevalence internalizing mental health symptoms. Additionally, we conducted Pearson correlations, separated by sex, to examine bivariate correlations amongst the physical (body fat percentage, BMI, physical activity) and mental health (stress, depression, anxiety, body image, self-esteem) dependent measures as well as possible sex differences in these relationships.”

The results of the t-tests are presented on pgs 6-7, section 3.1, lines 241-243: “Females exhibited significantly lower physical activity (t(228) = -4.35, p < .001) and greater anxiety (t(232) = 5.90, p < .001), depressive (t(233) = 5.35, p < .001), and stress symptoms (t(180) = 5.96, p < .001) compared to males.”

Additionally, the correlation analyses have been updated to assess correlations among males and females separately (pgs 8-9, section 3.2, lines 267-289):

“The bivariate correlations revealed significant positive correlations between body fat percentage and anxiety (rm = .22, pm = .04; rf = .19, pf = .04) among males and females and higher stress among males (r = .23, p = .02), higher body fat percentage was associated with more anxiety symptoms for males and females and higher stress for males. Additionally, body fat percentage was negatively correlated with body satisfaction for females (r = -.45, p = .02), but not males (r = -.26, p = .09), higher body fat percentage was associated with lower body satisfaction among females, but not males. BMI was positively correlated with anxiety (rm = .26, pm = .02; rf = .21, pf = .02) and depression (rm = .22, pm < .05; rf = .18, pf = .04) among males and females and stress among males (r = -.26, p = .03), higher BMI was associated with more anxiety and depressive symptoms among males and females and higher stress among males. BMI was also negatively correlated with body satisfaction (rm = -.31, pm = .02; rf = -.45, pf = .02) among males and females and self-esteem among females (r = -.32, p = .02), higher BMI was associated with lower body satisfaction among males and females and lower self-esteem among females. Body satisfaction was positively correlated with self-esteem (rm = .36, pm = .02; rf = .49, pf = .02;) and negatively correlated with depression (rm = -.31, pm = .04; rf = -.40, p f = .04) among males and females and anxiety among females (r = -.44, p = .02), higher body satisfaction was associated with greater self-esteem and less depressive symptoms among males and females and less anxiety symptoms among females. Self-esteem was also negatively correlated with anxiety (rm = -.44, pm < .01; rf = -.44, pf < .01), depression (rm = -.49, pm < .001; rf = -.71, pf < .001) and psychological stress (rm = -.39, pm < .01; rf = -.64, pf < .01), higher self-esteem was associated with less internalizing mental health symptoms for both males and females. Physical activity was positively correlated with anxiety (r = .18, p < .05) and depression (r = .24, p < .01) among females, but not males, indicating greater physical activity was associated with increased anxiety and depressive symptoms among females. Interestingly, neither body fat percentage (rm = -.03, pm = .78; rf = -.14, pf = .13) nor BMI (rm = -.10, pm = .35; rf = -.12, pf = .19) were significantly correlated with self-report physical activity levels among males or females. A correlation matrix containing relationships between all the examined variables for male participants is depicted in Table 4 and females are in Table 5.”

Section 4.2 of the discussion has been updated to reflect the updated analyses (pg 13, section 4.2, lines 359-371):

“The bivariate correlations revealed that higher body composition was associated with greater anxiety and depressive among males and females and greater stress among males. Relationships between body composition, body satisfaction, and self-esteem differed by sex, where body composition was more closely related to body satisfaction and self-esteem in females than males. Additionally, higher self-esteem was associated with less internalizing mental health symptoms among both males and females. Unexpectedly, there was a positive relationship between physical activity and anxiety depressive symptoms among females. The cause of this relationship is unknown and should be explored in future research. The present analysis demonstrated that relationships between body composition, body satisfaction, and mental health outcomes are different for males and females. This is in line with previous research which found that the relationships between body image and mental health outcomes were sex dependent; males that rated themselves as thin/very thin and females that rated themselves as very fat/chubby or thin/very thin had a higher risk of symptoms of anxiety or depression [15].”

  1. (Line 356 on page12) The physical activity in this study is self-reported PA, which indicates subjective PA. Therefore, the prevalence of PA in this study may be overestimated. An objective evaluation using an accelerometer is desirable. Please add this point. 

The following text has been added to pg 14, section 4.3, lines 417-419: “Further, a subjective, self-report measure of physical activity was used for the study, which may result in overestimation of physical activity. Future studies should strive to use an objective measure of physical activity such as accelerometry.”

Minor Comments

Title 

  1. The title misleads people into thinking that you have done a multi-level analysis. Please change it.

The title has been changed to read “Relationships between physical and mental health in adolescents from low-income, rural communities: Univariate and multivariate analyses”.

Method in Abstract

  1. Add that it was adjusted for gender in the correlation analysis.

The method in the abstract has been updated to reflect the separate correlations for males and females, the description now reads as follows: “Relationships between mental and physical health were assessed using Pearson correlations and multivariate data-driven cluster analysis. Positive correlations were observed between body composition and mental health symptoms, while negative correlations were observed between body image and mental health and body composition. However, sex differences were present in these relationships.”

Background

  1. (Line 83 on page2) Please add a brief explanation of the intersectionality theory.

This text has been changed to state: “Evidence regarding the impact of SES on mental health status has been mixed, possibly due to the fact that it may interact with factors, such as race, to create individualized experiences and perspectives. Indeed, intersectionality theory proposes that demographic factors, including sex, race/ethnicity, SES, and culture interact to form individualized experiences and perspectives which contribute to differences in mental health attitudes, access, and service use [20–22].”

Method

  1. (Line 206 on page6) Add that it was adjusted for gender in the correlation analysis.

The methods have been updated to reflect the separate correlations for males and females (pg 5, section 2.4, lines 201-204): “Additionally, we conducted Pearson correlations, separated by sex, to examine bivariate correlations amongst the physical (body fat percentage, BMI, physical activity) and mental health (stress, depression, anxiety, body image, self-esteem) dependent measures as well as possible sex differences in these relationships.”

Results

  1. (Line 238 on page6) The values are different from the table. Make the denominator consistent.

This change has been made.

Tables

  1. Spell out the abbreviations in the foot note: Table 1 HS GED; Table 3 PACER

These changes have been made.

Reviewer 2 Report

Thank you for giving me the opportunity to review this article, which aimed to quantify physical and mental health status among tenth and eleventh grade students attending four Title I high schools in rural Alabama. While reading the manuscript has been interesting and while the findings will make a contribution to the field, I think several aspects need to be clarified to facilitate for the reader. These have been outlined below.

Overarching/broader comments

  • Considering this article is part of a special issue focusing on improving the health of rural, minority populations and in order to increase the relevance of the study for the national – and especially the international – audience, it would benefit from including a section that clearly describes the setting of rural Alabama and the different schools included. This is also essential if the authors wish for the analysis to be replicated and extended in “similar populations” (line 347).

  • Overall, the conceptual clarity and consistency should be improved. Firstly, along with the authors first hypothesis, which indicate that this population would exhibit a high prevalence of physical and mental health problems compared to national standards, and the wish to examine relationships between physical and mental health, please make it clear throughout what kind of relationship you are investigating (i.e. cumulative ones). Secondly, mental health is a complex concept and only on one occasion do the authors state that they use internalizing symptoms as indicators of mental (ill)health. Please clarify earlier in the text that this is your explicit focus and remain consistent throughout with regard to the terminology. Thirdly, the broad term ‘environment’ is used to refer to some rural-urban disparity that remains largely unexplained. This is somewhat surprising given the focus of the study is rural Alabama (in and of itself) and not differences in physical and mental health between adolescents in rural as compared to urban parts of Alabama. Based on this notion, I suggest (and would commend the authors) if they placed a stronger emphasis on the rural context while allowing rurality to ‘stand apart’ from a largely normative urban comparison.

  • In the background, quite a lot of attention (lines 73-94) is paid to the issue of ethnicity/race, while less emphasis is placed on aspects like sex/gender. Based on this notions, the absent focus on ethnicity/race in the analysis comes as a slight surprise.

  • Please provide more details about the data collection. Specifically, how were the participants recruited, who collected the data and what was the response rate?

  • In section 3.2., please facilitate for the reader by interpreting and explaining what is implied the different positive or negative correlations such as, for example, “positive correlations between body fat percentage and anxiety (r = .20, p < .01), depression (r = .15, p = .03), stress (r = .19, p < .01), and BMI (r = .92, p < .001), indicating that a higher body fat percentage is associated with a higher BMI and more internalizing mental health symptoms”.

  • To my (very limited) knowledge about cluster analysis, as an analyst, you choose a solute based on a selection of the number of clusters; and while the authors write that their analysis revealed four distinct clusters of participants, it would be good to know if some alternative solutions were considered and rejected.

  • In the concluding section, the authors state that they identified “relationships between mental and physical health and the factors that influence those relationships at the group and individual levels”. Given they have only examined bivariate correlations between indicators of physical and mental health, I would say that factors which influence the identified relationships remain unexplored in this study.

  • The article is missing a discussion about ethics.

Specific/minor comments

  • The sentence referring to ref. 9 on lines 47-49, clarify whether the prevalence or the incidence has been estimated.

  • The sentences starting the third paragraph on page 2 (lines 69-76) seem to somewhat contradict the sentence ending the second paragraph (lines 65-68). Is there a lack of research? Or is there research that consistently find support for existing rural-urban disparities?

  • Throughout the text, please check and preferably limit applications of the terms i.e. and e.g.

Author Response

We appreciate the thoughtful reviews and believe that the manuscript is improved as a result. We have addressed the editors’ and each of the reviewers’ concerns below.

Thank you for giving me the opportunity to review this article, which aimed to quantify physical and mental health status among tenth and eleventh grade students attending four Title I high schools in rural Alabama. While reading the manuscript has been interesting and while the findings will make a contribution to the field, I think several aspects need to be clarified to facilitate for the reader. These have been outlined below.

Overarching/broader comments

  1. Considering this article is part of a special issue focusing on improving the health of rural, minority populations and in order to increase the relevance of the study for the national – and especially the international – audience, it would benefit from including a section that clearly describes the setting of rural Alabama and the different schools included. This is also essential if the authors wish for the analysis to be replicated and extended in “similar populations” (line 347).

Information regarding the rurality measures of the counties included in the study were added to pg 3, section 2.1, lines 116-120: “Both counties in which the high schools were located are considered rural, with an Index of Relative Rurality (IRR) of 0.5 (Chambers [IRR = 0.5]; Tallapoosa [IRR = 0.5]) in 2010 [26] Both counties are primarily White, with Tallapoosa county (70.0% White, 27.4% Black) having a higher percentage of White residents than Chambers county (57.7% White, 39.6% Black) [27].”

  1. Overall, the conceptual clarity and consistency should be improved. Firstly, along with the authors first hypothesis, which indicate that this population would exhibit a high prevalence of physical and mental health problems compared to national standards, and the wish to examine relationships between physical and mental health, please make it clear throughout what kind of relationship you are investigating (i.e. cumulative ones). Secondly, mental health is a complex concept and only on one occasion do the authors state that they use internalizing symptoms as indicators of mental (ill)health. Please clarify earlier in the text that this is your explicit focus and remain consistent throughout with regard to the terminology. Thirdly, the broad term ‘environment’ is used to refer to some rural-urban disparity that remains largely unexplained. This is somewhat surprising given the focus of the study is rural Alabama (in and of itself) and not differences in physical and mental health between adolescents in rural as compared to urban parts of Alabama. Based on this notion, I suggest (and would commend the authors) if they placed a stronger emphasis on the rural context while allowing rurality to ‘stand apart’ from a largely normative urban comparison.

The description of the hypothesis regarding relationships between physical and mental health has been changed to state (pg 3, section 1.1, lines 102-105): We hypothesized that this cumulative relationship may be moderated by sex, such that females would report lower physical activity and more mental health symptoms than males resulting in sex differences in the relationship between physical and mental health.

Changes were made throughout the text to clarify that we are discussing internalizing mental health issues.

Information regarding the rurality measures of the counties included in the study were added to pg 3, section 2.1, lines 116-1120: “Both counties in which the high schools were located are considered rural, with an Index of Relative Rurality (IRR) of 0.5 (Chambers [IRR = 0.5]; Tallapoosa [IRR = 0.5]) in 2010 [26] Both counties are primarily White, with Tallapoosa county (70.0% White, 27.4% Black) having a higher percentage of White residents than Chambers county (57.7% White, 39.6% Black) [27].”

  1. In the background, quite a lot of attention (lines 73-94) is paid to the issue of ethnicity/race, while less emphasis is placed on aspects like sex/gender. Based on this notions, the absent focus on ethnicity/race in the analysis comes as a slight surprise.

Much of the information regarding race/ethnicity (including these lines) have been removed from the introduction to focus more clearly on demonstrating the increased risk of the study population and the influence of sex on these risks.

Additionally, we have added independent t-tests to assess sex differences in physical activity and prevalence of internalizing mental health symptoms. Additionally, the correlation analyses have been changed to assess correlations among males and female separately to allow a more in-depth exploration of sex differences in these relationships. These changes to the statistical analyses are described on pg 5, section 2.4, lines 199-201:

“MATLAB version R2018a (MathWorks Inc., Natick, MA, USA) was used to conduct independent samples t-tests to examine sex differences in physical activity and prevalence internalizing mental health symptoms.”

The results of the t-tests are presented on pgs 6-7, section 3.1, lines 241-243: “Females exhibited significantly lower physical activity (t(228) = -4.35, p < .001) and greater anxiety (t(232) = 5.90, p < .001), depressive (t(233) = 5.35, p < .001), and stress symptoms (t(180) = 5.96, p < .001) compared to males.”

  1. Please provide more details about the data collection. Specifically, how were the participants recruited, who collected the data and what was the response rate?

The following text regarding recruitment and response rate has been added to pg 4, section 2.3, lines 177-179: “Information packets with details about the study were distributed to students 1-2 weeks prior to data collection. Out of 575 students, 253 students participated in the study (44%).”

The following text regarding who collected the data has been added to pg 5, section 2.3, lines 186-187: “Data were collected by trained research assistants during a single visit to each school (~3 hours total).”

  1. In section 3.2., please facilitate for the reader by interpreting and explaining what is implied the different positive or negative correlations such as, for example, “positive correlations between body fat percentage and anxiety (r = .20, p < .01), depression (r = .15, p = .03), stress (r = .19, p < .01), and BMI (r = .92, p < .001), indicating that a higher body fat percentage is associated with a higher BMI and more internalizing mental health symptoms”.

This section has been changed to state the following (pgs 8-9, section 3.2, lines 267-289):

“The bivariate correlations revealed significant positive correlations between body fat percentage and anxiety (rm = .22, pm = .04; rf = .19, pf = .04) among males and females and higher stress among males (r = .23, p = .02), higher body fat percentage was associated with more anxiety symptoms for males and females and higher stress for males. Additionally, body fat percentage was negatively correlated with body satisfaction for females (r = -.45, p = .02), but not males (r = -.26, p = .09), higher body fat percentage was associated with lower body satisfaction among females, but not males. BMI was positively correlated with anxiety (rm = .26, pm = .02; rf = .21, pf = .02) and depression (rm = .22, pm < .05; rf = .18, pf = .04) among males and females and stress among males (r = -.26, p = .03), higher BMI was associated with more anxiety and depressive symptoms among males and females and higher stress among males. BMI was also negatively correlated with body satisfaction (rm = -.31, pm = .02; rf = -.45, pf = .02) among males and females and self-esteem among females (r = -.32, p = .02), higher BMI was associated with lower body satisfaction among males and females and lower self-esteem among females. Body satisfaction was positively correlated with self-esteem (rm = .36, pm = .02; rf = .49, pf = .02;) and negatively correlated with depression (rm = -.31, pm = .04; rf = -.40, p f = .04) among males and females and anxiety among females (r = -.44, p = .02), higher body satisfaction was associated with greater self-esteem and less depressive symptoms among males and females and less anxiety symptoms among females. Self-esteem was also negatively correlated with anxiety (rm = -.44, pm < .01; rf = -.44, pf < .01), depression (rm = -.49, pm < .001; rf = -.71, pf < .001) and psychological stress (rm = -.39, pm < .01; rf = -.64, pf < .01), higher self-esteem was associated with less internalizing mental health symptoms for both males and females. Physical activity was positively correlated with anxiety (r = .18, p < .05) and depression (r = .24, p < .01) among females, but not males, indicating greater physical activity was associated with increased anxiety and depressive symptoms among females. Interestingly, neither body fat percentage (rm = -.03, pm = .78; rf = -.14, pf = .13) nor BMI (rm = -.10, pm = .35; rf = -.12, pf = .19) were significantly correlated with self-report physical activity levels among males or females. A correlation matrix containing relationships between all the examined variables for male participants is depicted in Table 4 and females are in Table 5.”

  1. To my (very limited) knowledge about cluster analysis, as an analyst, you choose a solute based on a selection of the number of clusters; and while the authors write that their analysis revealed four distinct clusters of participants, it would be good to know if some alternative solutions were considered and rejected.

We thank the reviewer for identifying the need to describe the selection of models and parameters for the cluster analysis. As a point of clarification, there are many different types of cluster analyses. Given no precedent for the use of cluster analysis to create profiles of individuals based on physical and mental health outcome measures, we did not want to constrain the cluster analysis to the hypothesized four clusters. In addition to clustering individuals (rows of our matrix), we were also interested in understanding the relationships between the variables themselves (columns of the matrix). For these reasons, we used a data-driven, divisive, hierarchical clustering approach using the clustergram function in MATLAB.

Using this method, the following parameters are defined:

  1. Dimension for standardizing data values (default: no standardization, however, we standardized participant data for each of the 13 variables due to differences in scale of values for these 13 variables)
  2. The function and threshold value for the measurement of distance between data (similarity/dissimilarity) for the rows and columns of data (default: Euclidian)
  3. The type of linkage function between pairs of observations (default: unweighted average distance).

Unlike k-means clustering, the number of clusters are not determined by the researcher, but rather, are based on the parameters above and the dataset.

Based on the reviewer’s question, we evaluated differences in the outcomes by changing the functions and threshold values for the parameters above. It is important to note that because we standardized our data to reduce differences in the scale of each of the 13 variables (and the influence of an individual variable to the clustering), differences in output were minimized between function and threshold values. Given these similarities, there was no reason to use values other than the default and what was presented in the present manuscript.

To provide validation of the number of clusters, we used R to run a cluster validation package called NbClust. This package evaluates 30 cluster indices and based on majority rule, the optimal number of clusters is identified. Four was identified as the optimal number of clusters, consistent with the results presented in the manuscript.

The following was added to the statistical analysis section (pg 5, section 2.4, lines 215-219):

“Unlike k-means clustering, the number of clusters is not predetermined in hierarchical divisive cluster analysis. Therefore, the number of clusters is based on the distance function, linkage function, and the structure of the dataset. However, variance in cluster output due to the selection of the distance function and linkage functions is minimized when data are normalized.”

The following was added as a note in the results section (3.4): “Note: the optimal number of clusters was examined using the NbClust package in R/R-Studio (version 1.2.1335). This package evaluates 30 cluster indices and based on majority rule, the optimal number of clusters is identified. Four was identified as the optimal number of clusters.”

  1. In the concluding section, the authors state that they identified “relationships between mental and physical health and the factors that influence those relationships at the group and individual levels”. Given they have only examined bivariate correlations between indicators of physical and mental health, I would say that factors which influence the identified relationships remain unexplored in this study.

This sentence has been changed to state: “Second, relationships between mental and physical health were identified at the group and individual levels in this population.”

  1. The article is missing a discussion about ethics.

Ethics information, including the IRB protocol is located on pg 4, section 2.3, lines 175-176: “The study was conducted in accordance with the Declaration of Helsinki, and the protocol was approved by the Institutional Review Board at Auburn University (Protocol #: 18-109 MR 1803).”

Specific/minor comments

  1. The sentence referring to ref. 9 on lines 47-49, clarify whether the prevalence or the incidence has been estimated.

This sentence refers to prevalence as stated in the text: “Both clinical prevalence of anxiety and depression as well as reported depressive symptoms are higher in females than males [2,7,8].”

  1. The sentences starting the third paragraph on page 2 (lines 69-76) seem to somewhat contradict the sentence ending the second paragraph (lines 65-68). Is there a lack of research? Or is there research that consistently find support for existing rural-urban disparities?

As stated in the text, there is a lack of research examining relationships between physical and mental health in this specific population (i.e., youth from rural, low-income communities). There is evidence for existing rural-urban disparities in physical activity and health, in which rural populations participate in less physical activity and are more likely to be overweight or obese (Johnson & Johnson, 2015).

  1. Throughout the text, please check and preferably limit applications of the terms i.e. and e.g.

Extraneous instances of “e.g.” and “i.e.” were removed throughout the text.

Round 2

Reviewer 1 Report

Minor comment

page 7 lines 246-247

The values in the text are still different from the values in Table 3. Please check them again.

Author Response

We would like to thank the reviewer for their continued efforts to improve the manuscript. The reviewer's comment is addressed below.

Minor Comments

1. page 7 lines 246-247- The values in the text are still different from the values in Table 3. Please check them again.

Response: The text on page 7, lines 246-247 has been changed to state: "Based on BMI measurements, 39.1% of participants were either overweight (15.8%) or obese (23.3%)." These percentages match those presented in table 3.